# Efficient Approximation Algorithms for String Kernel Based Sequence Classification

**Muhammad Farhan**
Department of Computer Science
School of Science and Engineering
Lahore University of Management Sciences
Lahore, Pakistan
14030031@lums.edu.pk

**Juvaria Tariq**
Department of Mathematics
School of Science and Engineering
Lahore University of Management Sciences
Lahore, Pakistan
jtariq@emory.edu

**Arif Zaman**
Department of Computer Science
School of Science and Engineering
Lahore University of Management Sciences
Lahore, Pakistan
arifz@lums.edu.pk

**Mudassir Shabbir**
Department of Computer Science
Information Technology University
Lahore, Pakistan
mudassir.shabbir@itu.edu.pk

**Imdad Ullah Khan**
Department of Computer Science
School of Science and Engineering
Lahore University of Management Sciences
Lahore, Pakistan
imdad.khan@lums.edu.pk

## Abstract

Sequence classification algorithms, such as SVM, require a definition of distance (similarity) measure between two sequences. A commonly used notion of similarity is the number of matches between $k$-mers ($k$-length subsequences) in the two sequences. Extending this definition, by considering two $k$-mers to match if their distance is at most $m$, yields better classification performance. This, however, makes the problem computationally much more complex. Known algorithms to compute this similarity have computational complexity that render them applicable only for small values of $k$ and $m$. In this work, we develop novel techniques to efficiently and accurately estimate the pairwise similarity score, which enables us to use much larger values of $k$ and $m$, and get higher predictive accuracy. This opens up a broad avenue of applying this classification approach to audio, images, and text sequences. Our algorithm achieves excellent approximation performance with theoretical guarantees. In the process we solve an open combinatorial problem, which was posed as a major hindrance to the scalability of existing solutions. We give analytical bounds on quality and runtime of our algorithm and report its empirical performance on real world biological and music sequences datasets.

## 1   Introduction

Sequence classification is a fundamental task in pattern recognition, machine learning, and data mining with numerous applications in bioinformatics, text mining, and natural language processing. Detecting proteins homology (shared ancestry measured from similarity of their sequences of amino

acids) and predicting proteins fold (functional three dimensional structure) are essential tasks in bioinformatics. Sequence classification algorithms have been applied to both of these problems with great success [3, 10, 13, 18, 19, 20, 25]. Music data, a real valued signal when discretized using vector quantization of MFCC features is another flavor of sequential data [26]. Sequence classification has been used for recognizing genres of music sequences with no annotation and identifying artists from albums [12, 13, 14]. Text documents can also be considered as sequences of words from a language lexicon. Categorizing texts into classes based on their topics is another application domain of sequence classification [11, 15].

While general purpose classification methods may be applicable to sequence classification, huge lengths of sequences, large alphabet sizes, and large scale datasets prove to be rather challenging for such techniques. Furthermore, we cannot directly apply classification algorithms devised for vectors in metric spaces because in almost all practical scenarios sequences have varying lengths unless some mapping is done beforehand. In one of the more successful approaches, the variable-length sequences are represented as fixed dimensional feature vectors. A feature vector typically is the spectra (counts) of all $k$-length substrings ($k$-mers) present exactly [18] or inexactly (with up to $m$ mismatches) [19] within a sequence. A *kernel function* is then defined that takes as input a pair of feature vectors and returns a real-valued similarity score between the pair (typically inner-product of the respective spectra's). The matrix of pairwise similarity scores (the kernel matrix) thus computed is used as input to a standard *support vector machine* (SVM) [5, 27] classifier resulting in excellent classification performance in many applications [19]. In this setting $k$ (the length of substrings used as bases of feature map) and $m$ (the mismatch parameter) are independent variables directly related to classification accuracy and time complexity of the algorithm. It has been established that using larger values of $k$ and $m$ improve classification performance [11, 13]. On the other hand, the runtime of kernel computation by the efficient trie-based algorithm [19, 24] is $O(k^{m+1}|\Sigma|^m(|X| + |Y|))$ for two sequences $X$ and $Y$ over alphabet $\Sigma$.

Computation of mismatch kernel between two sequences $X$ and $Y$ reduces to the following two problems. i) Given two $k$-mers $\alpha$ and $\beta$ that are at Hamming distance $d$ from each other, determine the size of intersection of $m$-mismatch neighborhoods of $\alpha$ and $\beta$ ($k$-mers that are at distance at most $m$ from both of them). ii) For $0 \leq d \leq \min\{2m, k\}$ determine the number of pairs of $k$-mers $(\alpha, \beta) \in X \times Y$ such that Hamming distance between $\alpha$ and $\beta$ is $d$. In the best known algorithm [13] the former problem is addressed by precomputing the intersection size in constant time for $m \leq 2$ only. While a sorting and enumeration based technique is proposed for the latter problem that has computational complexity $O(2^k(|X| + |Y|))$, which makes it applicable for moderately large values of $k$ (of course limited to $m \leq 2$ only).

In this paper, we completely resolve the combinatorial problem (problem i) for all values of $m$. We prove a closed form expression for the size of intersection of $m$-mismatch neighborhoods that lets us precompute these values in $O(m^3)$ time (independent of $|\Sigma|$, $k$, lengths and number of sequences). For the latter problem we devise an efficient approximation scheme inspired by the theory of locality sensitive hashing to accurately estimate the number of $k$-mer pairs between the two sequences that are at distance $d$. Combining the above two we design a polynomial time approximation algorithm for kernel computation. We provide probabilistic guarantees on the quality of our algorithm and analytical bounds on its runtime. Furthermore, we test our algorithm on several real world datasets with large values of $k$ and $m$ to demonstrate that we achieve excellent predictive performance. Note that string kernel based sequence classification was previously not feasible for this range of parameters.

## 2 Related Work

In the computational biology community pairwise alignment similarity scores were used traditionally as basis for classification, like the local and global alignment [5, 29]. String kernel based classification was introduced in [30, 9]. Extending this idea, [30] defined the *gappy $n$-gram kernel* and used it in conjunction with SVM [27] for text classification. The main drawback of this approach is that runtime for kernel evaluations depends quadratically on lengths of the sequences.

An alternative model of string kernels represents sequences as fixed dimensional vectors of counts of occurrences of $k$-mers in them. These include $k$-spectrum [18] and substring [28] kernels. This notion is extended to count inexact occurrences of patterns in sequences as in mismatch [19] and profile [10] kernels. In this transformed feature space SVM is used to learn class boundaries. This approach

yields excellent classification accuracies [13] but computational complexity of kernel evaluation remains a daunting challenge [11].

The exponential dimensions ($|\Sigma|^k$) of the feature space for both the $k$-spectrum kernel and $k, m$-mismatch kernel make explicit transformation of strings computationally prohibitive. SVM does not require the feature vectors explicitly; it only uses pairwise dot products between them. A trie-based strategy to implicitly compute kernel values for pairs of sequences was proposed in [18] and [19]. A $(k, m)$-mismatch tree is introduced which is a rooted $|\Sigma|$-ary tree of depth $k$, where each internal node has a child corresponding to each symbol in $\Sigma$ and every leaf corresponds to a $k$-mer in $\Sigma^k$. The runtime for computing the $k, m$ mismatch kernel value between two sequences $X$ and $Y$, under this trie-based framework, is $O((|X| + |Y|)k^{m+1}|\Sigma|^m)$, where $|X|$ and $|Y|$ are lengths of sequences. This makes the algorithm only feasible for small alphabet sizes and very small number of allowed mismatches.

The $k$-mer based kernel framework has been extended in several ways by defining different string kernels such as restricted gappy kernel, substitution kernel, wildcard kernel [20], cluster kernel [32], sparse spatial kernel [12], abstraction-augmented kernel [16], and generalized similarity kernel [14]. For literature on large scale kernel learning and kernel approximation see [34, 1, 7, 22, 23, 33] and references therein.

## 3 Algorithm for Kernel Computation

In this section we formulate the problem, describe our algorithm and analyze it's runtime and quality.

$k$-**spectrum and** $k, m$-**mismatch kernel:** Given a sequence $X$ over alphabet $\Sigma$, the $k, m$-mismatch spectrum of $X$ is a $|\Sigma|^k$-dimensional vector, $\Phi_{k,m}(X)$ of number of times each possible $k$-mer occurs in $X$ with at most $m$ mismatches. Formally,

$$\Phi_{k,m}(X) = (\Phi_{k,m}(X)[\gamma])_{\gamma \in \Sigma^k} = \left( \sum_{\alpha \in X} I_m(\alpha, \gamma) \right)_{\gamma \in \Sigma^k}, \qquad (1)$$

where $I_m(\alpha, \gamma) = 1$, if $\alpha$ belongs to the set of $k$-mers that differ from $\gamma$ by at most $m$ mismatches, i.e. the Hamming distance between $\alpha$ and $\gamma$, $d(\alpha, \gamma) \leq m$. Note that for $m = 0$, it is known as $k$-*spectrum* of $X$. The $k, m$-*mismatch* kernel value for two sequences $X$ and $Y$ (the mismatch spectrum similarity score) [19] is defined as:

$$K(X, Y | k, m) = \langle \Phi_{k,m}(X), \Phi_{k,m}(Y) \rangle = \sum_{\gamma \in \Sigma^k} \Phi_{k,m}(X)[\gamma] \Phi_{k,m}(Y)[\gamma]$$

$$= \sum_{\gamma \in \Sigma^k} \sum_{\alpha \in X} I_m(\alpha, \gamma) \sum_{\beta \in Y} I_m(\beta, \gamma) = \sum_{\alpha \in X} \sum_{\beta \in Y} \sum_{\gamma \in \Sigma^k} I_m(\alpha, \gamma) I_m(\beta, \gamma). \qquad (2)$$

For a $k$-mer $\alpha$, let $N_{k,m}(\alpha) = \{\gamma \in \Sigma^k : d(\alpha, \gamma) \leq m\}$ be the $m$-*mutational neighborhood* of $\alpha$. Then for a pair of sequences $X$ and $Y$, the $k, m$-mismatch kernel given in eq (2) can be equivalently computed as follows [13]:

$$K(X, Y | k, m) = \sum_{\alpha \in X} \sum_{\beta \in Y} \sum_{\gamma \in \Sigma^k} I_m(\alpha, \gamma) I_m(\beta, \gamma)$$

$$= \sum_{\alpha \in X} \sum_{\beta \in Y} |N_{k,m}(\alpha) \cap N_{k,m}(\beta)| = \sum_{\alpha \in X} \sum_{\beta \in Y} \mathfrak{I}_m(\alpha, \beta), \qquad (3)$$

where $\mathfrak{I}_m(\alpha, \beta) = |N_{k,m}(\alpha) \cap N_{k,m}(\beta)|$ is the size of intersection of $m$-mutational neighborhoods of $\alpha$ and $\beta$. We use the following two facts.

**Fact 3.1.** $\mathfrak{I}_m(\alpha, \beta)$, *the size of the intersection of $m$-mismatch neighborhoods of $\alpha$ and $\beta$, is a function of $k$, $m$, $|\Sigma|$ and $d(\alpha, \beta)$ and is independent of the actual $k$-mers $\alpha$ and $\beta$ or the actual positions where they differ. (See section 3.1)*

**Fact 3.2.** *If $d(\alpha, \beta) > 2m$, then $\mathfrak{I}_m(\alpha, \beta) = 0$.*

In view of the above two facts we can rewrite the kernel value (3) as

$$K(X, Y | k, m) = \sum_{\alpha \in X} \sum_{\beta \in Y} \mathfrak{I}_m(\alpha, \beta) = \sum_{i=0}^{\min\{2m,k\}} M_i \cdot \mathcal{I}_i, \qquad (4)$$

where $\mathcal{I}_i = \mathfrak{I}_m(\alpha, \beta)$ when $d(\alpha, \beta) = i$ and $M_i$ is the number of pairs of $k$-mers $(\alpha, \beta)$ such that $d(\alpha, \beta) = i$, where $\alpha \in X$ and $\beta \in Y$. Note that bounds on the last summation follows from Fact 3.2 and the fact that the Hamming distance between two $k$-mers is at most $k$. Hence the problem of kernel evaluation is reduced to computing $M_i$'s and evaluating $\mathcal{I}_i$'s.

## 3.1 Closed form for Intersection Size

Let $N_{k,m}(\alpha, \beta)$ be the intersection of $m$-mismatch neighborhoods of $\alpha$ and $\beta$ i.e.

$$N_{k,m}(\alpha, \beta) = N_{k,m}(\alpha) \cap N_{k,m}(\beta).$$

As defined earlier $|N_{k,m}(\alpha, \beta)| = \mathfrak{I}_m(\alpha, \beta)$. Let $N_q(\alpha) = \{\gamma \in \Sigma^k : d(\alpha, \gamma) = q\}$ be the set of $k$-mers that differ with $\alpha$ in exactly $q$ indices. Note that $N_q(\alpha) \cap N_r(\alpha) = \emptyset$ for all $q \neq r$. Using this and defining $n^{qr}(\alpha, \beta) = |N_q(\alpha) \cap N_r(\beta)|$,

$$N_{k,m}(\alpha, \beta) = \bigcup_{q=0}^{m} \bigcup_{r=0}^{m} N_q(\alpha) \cap N_r(\beta) \quad \text{and} \quad \mathfrak{I}_m(\alpha, \beta) = \sum_{q=0}^{m} \sum_{r=0}^{m} n^{qr}(\alpha, \beta).$$

Hence we give a formula to compute $n^{ij}(\alpha, \beta)$. Let $s = |\Sigma|$.

**Theorem 3.3.** *Given two $k$-mers $\alpha$ and $\beta$ such that $d(\alpha, \beta) = d$, we have that*

$$n^{ij}(\alpha, \beta) = \sum_{t=0}^{\frac{i+j-d}{2}} \binom{2d-i-j+2t}{d-(i-t)} \binom{d}{i+j-2t-d} (s-2)^{i+j-2t-d} \binom{k-d}{t} (s-1)^t.$$

*Proof.* $n^{ij}(\alpha, \beta)$ can be interpreted as the number of ways to make $i$ changes in $\alpha$ and $j$ changes in $\beta$ to get the same string. For clarity, we first deal with the case when we have $d(\alpha, \beta) = 0$, i.e both strings are identical. We wish to find $n^{ij}(\alpha, \beta) = |N_i(\alpha) \cap N_j(\beta)|$. It is clear that in this case $i = j$, otherwise making $i$ and $j$ changes to the same string will not result in the same string. Hence $n^{ij} = \binom{k}{i}(s-1)^i$. Second we consider $\alpha, \beta$ such that $d(\alpha, \beta) = k$. Clearly $k \geq i$ and $k \geq j$. Moreover, since both strings do not agree at any index, character at every index has to be changed in at least one of $\alpha$ or $\beta$. This gives $k \leq i + j$.

Now for a particular index $p$, $\alpha[p]$ and $\beta[p]$ can go through any one of the following three changes. Let $\alpha[p] = x$, $\beta[p] = y$. (I) Both $\alpha[p]$ and $\beta[p]$ may change from $x$ and $y$ respectively to some character $z$. Let $l_1$ be the count of indices going through this type of change. (II) $\alpha[p]$ changes from $x$ to $y$, call the count of these $l_2$. (III) $\beta[p]$ changes from $y$ to $x$, let this count be $l_3$. It follows that

$$i = l_1 + l_2 \quad , \quad j = l_1 + l_3, \quad , \quad l_1 + l_2 + l_3 = k.$$

This results in $l_1 = i + j - k$. Since $l_1$ is the count of indices at which characters of both strings change, we have $s - 2$ character choices for each such index and $\binom{k}{i+j-k}$ possible combinations of indices for $l_1$. From the remaining $l_2 + l_3 = 2k - i - j$ indices, we choose $l_2 = k - j$ indices in $\binom{2k-i-j}{k-j}$ ways and change the characters at these indices of $\alpha$ to characters of $\beta$ at respective indices. Finally, we are left with only $l_3$ remaining indices and we change them according to the definition of $l_3$. Thus the total number of strings we get after making $i$ changes in $\alpha$ and $j$ changes in $\beta$ is

$$(s-2)^{i+j-k} \binom{k}{i+j-k} \binom{2k-i-j}{k-j}.$$

Now we consider general strings $\alpha$ and $\beta$ of length $k$ with $d(\alpha, \beta) = d$. Without loss of generality assume that they differ in the first $d$ indices. We parameterize the system in terms of the number of changes that occur in the last $k - d$ indices of the strings i.e let $t$ be the number of indices that go through a change in last $k - d$ indices. Number of possible such changes is

$$\binom{k-d}{t}(s-1)^t. \tag{5}$$

Lets call the first $d$-length substrings of both strings $\alpha'$ and $\beta'$. There are $i - t$ characters to be changed in $\alpha'$ and $j - t$ in $\beta'$. As reasoned above, we have $d \leq (i-t) + (j-t) \implies t \leq \frac{i+j-d}{2}$.

In this setup we get $i - t = l_1 + l_2$, $j - t = l_1 + l_3$, $l_1 + l_2 + l_3 = d$ and $l_1 = (i-t) + (j-t) - d$. We immediately get that for a fixed $t$, the total number of resultant strings after making $i - t$ changes in $\alpha'$ and $j - t$ changes in $\beta'$ is

$$\binom{2d - (i-t) - (j-t)}{d - (i-t)} \binom{d}{(i-t) + (j-t) - d} (s-2)^{(i-t)+(j-t)-d}. \tag{6}$$

For a fixed $t$, every substring counted in (5), every substring counted in (6) gives a required string obtained after $i$ and $j$ changes in $\alpha$ and $\beta$ respectively. The statement of the theorem follows. $\square$

**Corollary 3.4.** *Runtime of computing $\mathcal{I}_d$ is $O(m^3)$, independent of $k$ and $|\Sigma|$.*

This is so, because if $d(\alpha, \beta) = d$, $\mathcal{I}_d = \sum_{q=0}^{m} \sum_{r=0}^{m} n^{qr}(\alpha, \beta)$ and $n^{qr}(\alpha, \beta)$ can be computed in $O(m)$.

## 3.2 Computing $M_i$

Recall that given two sequences $X$ and $Y$, $M_i$ is the number of pairs of $k$-mers $(\alpha, \beta)$ such that $d(\alpha, \beta) = i$, where $\alpha \in X$ and $\beta \in Y$. Formally, the problem of computing $M_i$ is as follows:

**Problem 3.5.** *Given $k$, $m$, and two sets of $k$-mers $S_X$ and $S_Y$ (set of $k$-mers extracted from the sequences $X$ and $Y$ respectively) with $|S_X| = n_X$ and $|S_Y| = n_Y$. Compute*

$$M_i = |\{(\alpha, \beta) \in S_X \times S_Y : d(\alpha, \beta) = i\}| \quad \text{for } 0 \leq i \leq \min\{2m, k\}.$$

Note that the brute force approach to compute $M_i$ requires $O(n_X \cdot n_Y \cdot k)$ comparisons. Let $\mathcal{Q}_k(j)$ denote the set of all $j$-sets of $\{1, \ldots, k\}$ (subsets of indices). For $\theta \in \mathcal{Q}_k(j)$ and a $k$-mer $\alpha$, let $\alpha|_\theta$ be the $j$-mer obtained by selecting the characters at the $j$ indices in $\theta$. Let $f_\theta(X, Y)$ be the number of pairs of $k$-mers in $S_X \times S_Y$ as follows;

$$f_\theta(X, Y) = |\{(\alpha, \beta) \in S_X \times S_Y : d(\alpha|_\theta, \beta|_\theta) = 0\}|.$$

We use the following important observations about $f_\theta$.

**Fact 3.6.** *For $0 \leq i \leq k$ and $\theta \in \mathcal{Q}_k(k-i)$, if $d(\alpha|_\theta, \beta|_\theta) = 0$, then $d(\alpha, \beta) \leq i$.*

**Fact 3.7.** *For $0 \leq i \leq k$ and $\theta \in \mathcal{Q}_k(k-i)$, $f_\theta(X, Y)$ can be computed in $O(kn \log n)$ time.*

This can be done by first lexicographically sorting the $k$-mers in each of $S_X$ and $S_Y$ by the indices in $\theta$. The pairs in $S_X \times S_Y$ that are the same at indices in $\theta$ can then be enumerated in one linear scan over the sorted lists. Let $n = n_X + n_Y$, runtime of this computation is $O(k(n + |\Sigma|))$ if we use counting sort (as in [13]) or $O(kn \log n)$ for mergesort (since $\theta$ has $O(k)$ indices.) Since this procedure is repeated many times, we refer to this as the SORT-ENUMERATE subroutine. We define

$$F_i(X, Y) = \sum_{\theta \in \mathcal{Q}_k(k-i)} f_\theta(X, Y). \tag{7}$$

**Lemma 3.8.**

$$F_i(X, Y) = \sum_{j=0}^{i} \binom{k-j}{k-i} M_j. \tag{8}$$

*Proof.* Let $(\alpha, \beta)$ be a pair that contributes to $M_j$, i.e. $d(\alpha, \beta) = j$. Then for every $\theta \in \mathcal{Q}_k(k-i)$ that has all indices within the $k - j$ positions where $\alpha$ and $\beta$ agree, the pair $(\alpha, \beta)$ is counted in $f_\theta(X, Y)$. The number of such $\theta$'s are $\binom{k-j}{k-i}$, hence $M_j$ is counted $\binom{k-j}{k-i}$ times in $F_i(X, Y)$, yielding the required equality. $\square$

**Corollary 3.9.** *$M_i$ can readily be computed as: $M_i = F_i(X, Y) - \sum_{j=0}^{i-1} \binom{k-j}{k-i} M_j$.*

By definition, $F_i(X, Y)$ can be computed with $\binom{k}{k-i} = \binom{k}{i}$ $f_\theta$ computations. Let $t = \min\{2m, k\}$. $K(X, Y|k, m)$ can be evaluated by (4) after computing $M_i$ (by (8)) and $\mathcal{I}_i$ (by Corollary 3.4) for $0 \leq i \leq t$. The overall complexity of this strategy thus is

$$\left( \sum_{i=0}^{t} \binom{k}{i} (k-i)(n \log n + n) \right) + O(n) = O(k \cdot 2^{k-1} \cdot (n \log n)).$$

---

**Algorithm 1** : Approximate-Kernel($S_X$,$S_Y$,$k$,$m$,$\epsilon$,$\delta$,$B$)

---

1: $\mathcal{I}, M' \leftarrow$ ZEROS($t+1$)
2: $\sigma \leftarrow \epsilon \cdot \sqrt{\delta}$
3: Populate $\mathcal{I}$ using Corollary 3.4
4: **for** $i = 0$ to $t$ **do**
5:      $\mu_F \leftarrow 0$
6:      $iter \leftarrow 1$
7:      $var_F \leftarrow \infty$
8:      **while** $var_F > \sigma^2 \wedge iter < B$ **do**
9:          $\theta \leftarrow$ RANDOM($\binom{k}{k-i}$)
10:          $\mu_F \leftarrow \dfrac{\mu_F \cdot (iter - 1) + \text{SORT-ENUMERATE}(S_X, S_Y, k, \theta)}{iter}$     ▷ Application of Fact 3.7
11:          $var_F \leftarrow$ VARIANCE($\mu_F, var_F, iter$)     ▷ Compute online variance
12:          $iter \leftarrow iter + 1$
13:      $F'[i] \leftarrow \mu_F \cdot \binom{k}{k-i}$
14:      $M'[i] \leftarrow F'[i]$
15:      **for** $j = 0$ to $i - 1$ **do**     ▷ Application of Corollary 3.9
16:          $M'[i] \leftarrow M'[i] - \binom{k-j}{k-i} \cdot M'[j]$
17: $K' \leftarrow$ SUMPRODUCT($M', \mathcal{I}$)     ▷ Applying Equation (4)
18: **return** $K'$

---

We give our algorithm to approximate $K(X, Y|k, m)$, it's explanation followed by it's analysis.

Algorithm 1 takes $\epsilon, \delta \in (0, 1)$, and $B \in \mathbb{Z}^+$ as input parameters; the first two controls the accuracy of estimate while $B$ is an upper bound on the sample size. We use (7) to estimate $F_i = F_i(X, Y)$ with an online sampling algorithm, where we choose $\theta \in \mathcal{Q}_k(k - i)$ uniformly at random and compute the online mean and variance of the estimate for $F_i$. We continue to sample until the variance is below the threshold ($\sigma^2 = \epsilon^2 \delta$) or the sample size reaches the upper bound $B$. We scale up our estimate by the population size and use it to compute $M'_i$ (estimates of $M_i$) using Corollary 3.9. These $M'_i$ 's together with the precomputed exact values of $\mathcal{I}_i$'s are used to compute our estimate, $K'(X, Y|k, m, \sigma, \delta, B)$, for the kernel value using (4). First we give an analytical bound on the runtime of Algorithm 1 then we provide guarantees on it's performance.

**Theorem 3.10.** *Runtime of Algorithm 1 is bounded above by $O(k^2 n \log n)$.*

*Proof.* Observe that throughout the execution of the algorithm there are at most $tB$ computations of $f_\theta$, which by Fact 3.7 needs $O(kn \log n)$ time. Since $B$ is an absolute constant and $t \leq k$, we get that the total runtime of the algorithm is $O(k^2 n \log n)$. Note that in practice the while loop in line 8 is rarely executed for $B$ iterations; the deviation is within the desired range much earlier. $\square$

Let $K' = K'(X, Y|k, m, \epsilon, \delta, B)$ be our estimate (output of Algorithm 1) for $K = K(X, Y|k, m)$.

**Theorem 3.11.** *$K'$ is an unbiased estimator of the true kernel value, i.e. $E(K') = K$.*

*Proof.* For this we need the following result, whose proof is deferred.

**Lemma 3.12.** *$E(M'_i) = M_i$.*

By Line 17 of Algorithm 1, $E(K') = E(\sum_{i=0}^{t} \mathcal{I}_i M'_i)$. Using the fact that $\mathcal{I}_i$'s are constants and Lemma 3.12 we get that

$$E(K') = \sum_{i=0}^{t} \mathcal{I}_i E(M'_i) = \sum_{i=0}^{\min\{2m,k\}} \mathcal{I}_i M_i = K.$$

$\square$

**Theorem 3.13.** *For any $0 < \epsilon, \delta < 1$, Algorithm 1 is an $(\epsilon \mathcal{I}_{max}, \delta)-$additive approximation algorithm, i.e. $Pr(|K - K'| \geq \epsilon \mathcal{I}_{max}) < \delta$, where $\mathcal{I}_{max} = \max_i\{\mathcal{I}_i\}$.*

Note that these are very loose bounds, in practice we get approximation far better than these bounds. Furthermore, though $\mathcal{I}_{max}$ could be large, but it is only a fraction of one of the terms in summation for the kernel value $K(X, Y | k, m)$.

*Proof.* Let $F'_i$ be our estimate for $F_i(X, Y) = F_i$. We use the following bound on the variance of $K'$ that is proved later.

**Lemma 3.14.** $Var(K') \leq \delta(\epsilon \cdot \mathcal{I}_{max})^2$.

By Lemma 3.12 we have $E(K') = K$, hence by Lemma 3.14, $Pr[\|K' - K\|] \geq \epsilon \mathcal{I}_{max}$ is equivalent to $Pr[\|K' - E(K')\|] \geq \frac{1}{\sqrt{\delta}} \sqrt{Var(K')}$. By the Chebychev's inequality, this latter probability is at most $\delta$. Therefore, Algorithm 1 is an $(\epsilon \mathcal{I}_{max}, \delta)-$additive approximation algorithm. $\qquad\square$

*Proof.* (Proof of Lemma 3.12) We prove it by induction on $i$. The base case $(i = 0)$ is true as we compute $M'[0]$ exactly, i.e. $M'[0] = M[0]$. Suppose $E(M'_j) = M_j$ for $0 \leq j \leq i - 1$. Let $iter$ be the number of iterations for $i$, after execution of Line 10 we get

$$F'[i] = \mu_F \binom{k}{k-i} = \frac{\sum_{r=1}^{iter} f_{\theta_r}(X, Y)}{iter} \binom{k}{k-i},$$

where $\theta_r$ is the random $(k-i)$-set chosen in the $r$th iteration of the while loop. Since $\theta_r$ is chosen uniformly at random we get that

$$E(F'[i]) = E(\mu_F)\binom{k}{k-i} = E(f_{\theta_r}(X, Y))\binom{k}{k-i} = \frac{F_i(X, Y)}{\binom{k}{k-i}}\binom{k}{k-i}. \qquad (9)$$

After the loop on Line 15 is executed we get that $E(M'[i]) = F_i(X, Y) - \sum_{j=0}^{i-1}\binom{k-j}{k-i}E(M'_j)$. Using $E(M'_j) = M_j$ (inductive hypothesis) in (8) we get that $E(M'_i) = M_i$. $\qquad\square$

*Proof.* (Proof of Lemma 3.14) After execution of the while loop in Algorithm 1, we have $F'_i = \sum_{j=0}^{i}\binom{k-j}{k-i}M'_j$. We use the following fact that follows from basic calculations.

**Fact 3.15.** *Suppose* $X_0, \ldots, X_t$ *are random variables and let* $S = \sum_{i=0}^{t} a_i X_i$, *where* $a_0, \ldots, a_t$ *are constants. Then*

$$Var(S) = \sum_{i=0}^{t} a_i^2 Var(X_i) + 2\sum_{i=0}^{t}\sum_{j=i+1}^{t} a_i a_j Cov(X_i, X_j).$$

Using fact 3.15 and definitions of $\mathcal{I}_{max}$ and $\sigma$ we get that

$$Var(K') = \sum_{i=0}^{t}\mathcal{I}_i^2 Var(M'_i) + 2\sum_{i=0}^{t}\sum_{j=i+1}^{t}\mathcal{I}_i\mathcal{I}_j Cov(M'_i, M'_j)$$

$$\leq \mathcal{I}_{max}^2\left[\sum_{i=0}^{t} Var(M'_i) + 2\sum_{i=0}^{t}\sum_{j=i+1}^{t} Cov(M'_i, M'_j)\right] \leq \mathcal{I}_{max}^2 Var(F'_t) \leq \mathcal{I}_{max}^2\sigma^2 = \delta(\epsilon \cdot \mathcal{I}_{max})^2.$$

The last inequality follows from the following relation derived from definition of $F'_i$ and Fact 3.15.

$$Var(F'_t) = \sum_{i=0}^{t}\binom{k-i}{k-t}^2 Var(M'_i) + 2\sum_{i=0}^{t}\sum_{j=i+1}^{t}\binom{k-i}{k-t}\binom{k-j}{k-t}Cov(M'_i, M'_j). \qquad (10)$$

$\qquad\square$

# 4 Evaluation

We study the performance of our algorithm in terms of runtime, quality of kernel estimates and predictive accuracies on standard benchmark sequences datasets (Table 1) . For the range of parameters feasible for existing solutions, we generated kernel matrices both by algorithm of [13] (exact) and our algorithm (approximate). These experiments are performed on an Intel Xeon machine with (8 Cores, 2.1 GHz and 32 GB RAM) using the same experimental settings as in [13, 15, 17]. Since our algorithm is applicable for significantly wider range of $k$ and $m$, we also report classification performance with large $k$ and $m$. For our algorithm we used $B \in \{300, 500\}$ and $\sigma \in \{0.25, 0.5\}$ with no significant difference in results as implied by the theoretical analysis. In all reported results $B = 300$ and $\sigma = 0.5$. In order to perform comparisons, for a few combinations of parameters we generated exact kernel matrices of each dataset on a much more powerful machine (a cluster of 20 nodes, each having 24 CPU's with 2.5 GHz speed and $128GB$ RAM). Sources for datasets and source code are available at [1].

Table 1: Datasets description

| Name | Task | Classes | Seq. | Av.Len. | Evaluation |
|------|------|---------|------|---------|------------|
| Ding-Dubchak [6] | protein fold recognition | 27 | 694 | 169 | 10-fold CV |
| SCOP [4, 31] | protein homology detection | 54 | 7329 | 308 | 54 binary class. |
| Music [21, 26] | music genre recognition | 10 | 1000 | 2368 | 5-fold CV |
| Artist20 [8, 17] | artist identification | 20 | 1413 | 9854 | 6-fold CV |
| ISMIR [17] | music genre recognition | 6 | 729 | 10137 | 5-fold CV |

***Running Times:*** We report difference in running times for kernels generation in Figure 1. Exact kernels are generated using code provided by authors of [13, 14] for $8 \leq k \leq 16$ and $m = 2$ only. We achieve significant speedups for large values of $k$ (for $k = 16$ we get one order of magnitude gains in computational efficiency on all datasets). The running times for these algorithms are $O(2^k n)$ and $O(k^2 n \log n)$ respectively. We can use larger values of $k$ without an exponential penalty, which is visible in the fact that in all graphs, as $k$ increases the growth of running time of the exact algorithm is linear (on the log-scale), while that of our algorithm tends to taper off.

Figure 1: Log scaled plot of running time of approximate and exact kernel generation for $m = 2$

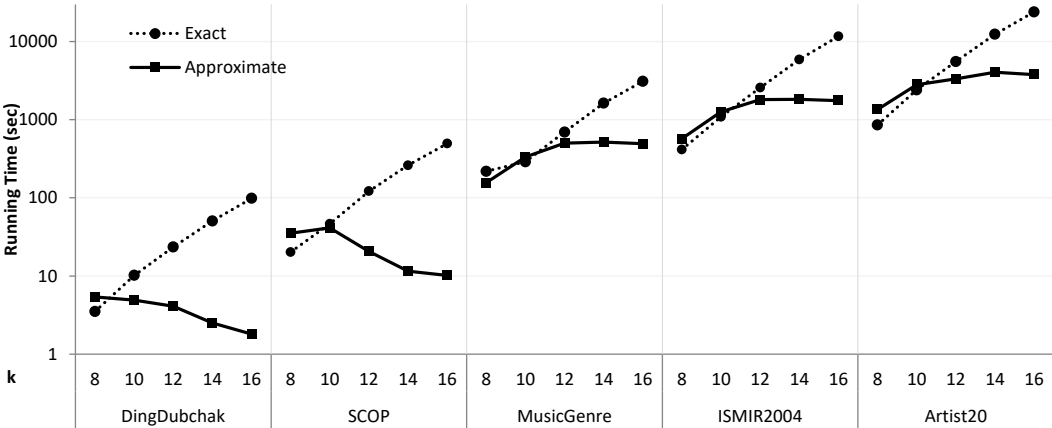

***Kernel Error Analysis:*** We show that despite reduction in runtimes, we get excellent approximation of kernel matrices. In Table 2 we report point-to-point error analysis of the approximate kernel matrices. We compare our estimates with exact kernels for $m = 2$. For $m > 2$ we report statistical error analyses. More precisely, we evaluate differences with principal submatrices of the exact kernel matrix. These principal submatrices are selected by randomly sampling 50 sequences and computing their pairwise kernel values. We report errors for four datasets; the fifth one, not included for space reasons, showed no difference in error. From Table 2 it is evident that our empirical performance is significantly more precise than the theoretical bounds proved on errors in our estimates.

Table 2: Mean absolute error (MAE) and root mean squared error (RMSE) of approximate kernels. For $m > 2$ we report average MAE and RMSE of three random principal submatrices of size $50 \times 50$

| $(k, m)$ | Music Genre | | ISMIR | | Artist20 | | SCOP | |
| --- | --- | --- | --- | --- | --- | --- | --- | --- |
| | RMSE | MAE | RMSE | MAE | RMSE | MAE | RMSE | MAE |
| $(10, 2)$ | 0 | 0 | 0 | 0 | 0 | 0 | 1.3E−6 | 9.0E−8 |
| $(12, 2)$ | 0 | 0 | 0 | 0 | 0 | 0 | 1.4E−6 | 1.0E−8 |
| $(14, 2)$ | 2.0E−8 | 0 | 2.0E−8 | 0 | 3.3E−8 | 1.3E−8 | 2.9E−6 | 1.3E−8 |
| $(16, 2)$ | 1.3E−8 | 0 | 4.0E−8 | 3.3E−9 | | | 2.9E−6 | 1.0E−8 |
| $(12, 6)$ | 1.97E−5 | 8.5E−7 | | | | | 2.4E−4 | 1.8E−5 |

***Prediction Accuracies:*** We compare the outputs of SVM on the exact and approximate kernels using the publicly available SVM implementation LIBSVM [2]. We computed exact kernel matrices by brute force algorithm for a few combinations of parameters for each dataset on the much more powerful machine. Generating these kernels took days; we only generated to compare classification performance of our algorithm with the exact one. We demonstrate that our predictive accuracies are sufficiently close to that with exact kernels in Table 3 (bio-sequences) and Table 4 (music). The parameters used for reporting classification performance are chosen in order to maintain comparability with previous studies. Similarly all measurements are made as in [13, 14], for instance for music genre classification we report results of 10-fold cross-validation (see Table 1). For our algorithm we used $B = 300$ and $\sigma = 0.5$ and we take an average of performances over three independent runs.

Table 3: Classification performance comparisons on SCOP (ROC) and Ding-Dubchak (Accuracy)

| | SCOP | | | | Ding-Dubchak | |
| --- | --- | --- | --- | --- | --- | --- |
| | Exact | | Approx | | Exact | Approx |
| $k, m$ | ROC | ROC50 | ROC | ROC50 | Accuracy | |
| 8, 2 | 88.09 | 38.71 | 88.05 | 38.60 | 34.01 | 31.65 |
| 10, 2 | 81.65 | 28.18 | 80.56 | 26.72 | 28.1 | 26.9 |
| 12, 2 | 71.31 | 23.27 | 66.93 | 11.04 | 27.23 | 26.66 |
| 14, 2 | 67.91 | 7.78 | 63.67 | 6.66 | 25.5 | 25.5 |
| 16, 2 | 64.45 | 6.89 | 61.64 | 5.76 | 25.94 | 25.03 |
| 10, 5 | 91.60 | 53.77 | 91.67 | 54.1 | 45.1 | 43.80 |
| 10, 7 | 90.27 | 48.18 | 90.30 | 48.44 | 58.21 | 57.20 |
| 12, 8 | 91.44 | 50.54 | 90.97 | 52.08 | 58.21 | 57.83 |

Table 4: Classification error comparisons on music datasets exact and estimated kernels

| $k, m$ | Music Genre | | ISMIR | | Artist20 | |
| --- | --- | --- | --- | --- | --- | --- |
| | Exact | Estimate | Exact | Estimate | Exact | Estimate |
| 10, 2 | $61.30 \pm 3.3$ | $61.30 \pm 3.3$ | $54.32 \pm 1.6$ | $54.32 \pm 1.6$ | $82.10 \pm 2.2$ | $82.10 \pm 2.2$ |
| 14, 2 | $71.70 \pm 3.0$ | $71.70 \pm 3.0$ | $55.14 \pm 1.1$ | $55.14 \pm 1.1$ | $86.84 \pm 1.8$ | $86.84 \pm 1.8$ |
| 16, 2 | $73.90 \pm 1.9$ | $73.90 \pm 1.9$ | $54.73 \pm 1.5$ | $54.73 \pm 1.5$ | $87.56 \pm 1.8$ | $87.56 \pm 1.8$ |
| 10, 7 | $37.00 \pm 3.5$ | $37.00 \pm 3.5$ | | $27.16 \pm 1.6$ | $55.75 \pm 4.7$ | $55.75 \pm 4.7$ |
| 12, 6 | $54.20 \pm 2.7$ | $54.13 \pm 2.9$ | $52.12 \pm 2.0$ | $52.08 \pm 1.5$ | $79.57 \pm 2.4$ | $80.00 \pm 2.6$ |
| 12, 8 | $43.70 \pm 3.2$ | $44.20 \pm 3.2$ | $47.03 \pm 2.6$ | $47.41 \pm 2.4$ | | $67.57 \pm 3.6$ |

## 5 Conclusion

In this work we devised an efficient algorithm for evaluation of string kernels based on inexact matching of subsequences ($k$-mers). We derived a closed form expression for the size of intersection of $m$-mismatch neighborhoods of two $k$-mers. Another significant contribution of this work is a novel statistical estimate of the number of $k$-mer pairs at a fixed distance between two sequences. Although large values of the parameters $k$ and $m$ were known to yield better classification results, known algorithms are not feasible even for moderately large values. Using the two above mentioned results our algorithm efficiently approximate kernel matrices with probabilistic bounds on the accuracy. Evaluation on several challenging benchmark datasets for large $k$ and $m$, show that we achieve state of the art classification performance, with an order of magnitude speedup over existing solutions.

## Footnotes

[1] `https://github.com/mufarhan/sequence_class_NIPS_2017`

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
