[Reviews · NeurIPS 2017]

Reviewer 1



This paper studies fast approximation of "mismatch string kernel" proposed by Leslie et al. [17], in that kernel distance between two strings are defined as the number of matches between k-mers while allowing up to m-mismaches. Since the original algorithm has a high computational complexity, the authors propose an approximation algorithm. It consists of two parts; (1) determine the size of m-mismatch neighbourhoods of two k-mers, and (2) determine the number of k-mer paris within Hamming distance d. For the former problem (1), the authors propose to precompute all m-mismatch neighbours with an O(m^3) algorithm, and for the latter problem (2), locality sensitive hashing is applied. The proposed algorithm is equipped with a probabilistic guarantees on the quality of the solution, and an analytical bounds on its running time. In computational experiments using both simulated and synthetic data, the proposed algorithm is shown to approximate the exact computation well, and run much faster especially when m is large. The paper is written clearly. The proposed algorithm is presented step by step with essential proofs. MINOR COMMENTS As is also mentioned by the authors, but proposed bound (Theorem 3.13) is loose. Isn't it non-trivial to improve it ? Some sentences in the section 4 are incomplete.

Reviewer 2



Paper summary: The authors propose an algorithm for the efficient estimation of the mismatch string kernel. They show empirically that the resulting estimate is accurate and provide theoretical guarantees to support their work. Moreover, empirical results support that using the kernel estimate does not hinder the accuracy of the SVM. Finally, the authors provide an upper bound on the runtime of their algorithm. Strengths and weaknesses: The paper is clear and well written. Proofs and technical parts of the paper are relatively easy to follow thanks to a consistent notation. It is great that the datasets used are from different domains and that the authors will make their source code available. This work is strongly based on the hypothesis that increasing the number of mismatch in the kernel will improve classification accuracy. Because of this, it would be great if this hypothesis was more strongly supported in the paper. In Figure 1, datasets are ordered by sequence lengths. We observe that, as sequence lengths increase, the running time benefit of the proposed approach diminish. Can the authors comment on this? Clarity & Quality: What are the values used for epsilon, delta and B in the Evaluation section? Additional experiments exploring the impact of these parameters would be a great addition to the paper. Why all five datasets are not part of Table 2, 3, and 4? If more space is needed, the authors can provide the complete result as supplementary material. Significance & Originality: The paper addresses a very specific question that will most likely appeal to narrow audience of NIPS. However, the work is of high quality and does succeed in improving the computational time of the mismatch kernel. Because the authors will make their source code available after publication, I believe that this paper can have a significant impact in its field. Errors / typos: Line 95: "belongs to theset of" Table 1: Are the evaluations column for the SCOP and Artist20 datasets correct? Line 193: Figure 4 should point to Figure 1 Line 204: "Generating these kernels days" Line 214: "an efficient for evaluation of"

Reviewer 3



The authors describe an approximation algorithm for k-mer (with mismatches) based string kernels. The contribution is centered around a closed form expression of the intersection size of mismatching neighbourhoods. The algorithm is evaluated in the context of sequence classification using SVM. I think this is a great paper: clear motivation, good introduction, clear contribution, theoretical back-up, nice experimental results. I have a few concerns regarding presentation and structuring, as well as doubts on relevance on the presented theory. Presentation. I think Section 3 is too technical. It contains a lot of notation, and a lot of clutter that actually hinder understanding the main ideas. On the other hand, intuition on crucial ideas is missing, so I suggest to move a few details into Appendix and rather elaborate high-level ideas. -A table with all relevant quantities (and explanations) would make it much easier for a non-string-kernel person to follow establishing Theorem 3.3. An intuitive description of the theorem (and proof idea) would also help. -Algorithm 1 is not really referenced from the main test. It first appears in Thm 3.10. It is also very sparsely annotated/explained. For example, the parameter meanings are not explained. Subroutines also need annotation. I think further deserves to be in the center of Section 3's text, and the text should describe the high level ideas. -It should be pointed out more that Thm 3.3 is the open combinatoric problem mentioned in the abstract. -In the interest of readability, I think the authors should comment more on what it is that their algorithm approximates, what the parameters that control the approximation quality are, and what would be possible failure cases. -Theorem 3.13 should stay in the main text as this concentration inequality is the main theoretical message. All results establishing the Chebbyshev inequality (Thm 3.11, Lemma 3.12, Lemma 3.14 requirements should go to the appendix. Rather, the implications of the Theorem 3.13 should be elaborated on more. I appreciate that the authors note that these bounds are extremely loose. Theory. -Theorem 3.13 is a good first step to understanding the approximation quality (and indeed consistency) of the algorithm. It is, however, not useful in the sense that we do not care about the kernel approximation itself, but we care about the generalization error in the downstream algorithm. Kernel method theory has seen a substantial amount of work in that respect (e.g. for Nystrom, Fourier Feature regression/classification). This should be acknowledged, or even better: established. A simple approach would be perturbation analysis using the established kernel approximation error, better would be directly controlling the error of the estimated decision boundary of the SVM. Experiments. -Runtime should be checked for various m -Where does the approximation fail? An instructive synthetic example would help understanding when the algorithm is appropriate and when it is not. -The authors mention that their algorithm allows for previously impossible settings of (k,m). In Table 3, however, there is only a single case where they demonstrate an improved performance as opposed to the exact algorithm (and the improvement is marginal). Either the authors need to exemplify the statement that their algorithm allows to solve previously unsolved problems (or allow for better accuracy), or they should remove it. Minor. -Figure 1 has no axis units -Typo line 95 "theset" -Typo line 204 "Generating these kernels days;"